# Revealing the Increased Stress Response Behavior through Transcriptomic Analysis of Adult Zebrafish Brain after Chronic Low to Moderate Dose Rates of Ionizing Radiation

**DOI:** 10.3390/cancers14153793

**Published:** 2022-08-04

**Authors:** Elsa Cantabella, Virginie Camilleri, Isabelle Cavalie, Nicolas Dubourg, Béatrice Gagnaire, Thierry D. Charlier, Christelle Adam-Guillermin, Xavier Cousin, Oliver Armant

**Affiliations:** 1Institut de Radioprotection et de Sûreté Nucléaire (IRSN), Pôle Santé Environnement-Environnement (PSE-ENV)/Service de Recherche sur les Transferts et les Effets des Radionucléides sur les Ecosystèmes (SRTE)/Laboratoire de Recherche sur les Effets des Radionucléides sur les Ecosystèmes (LECO), Cadarache, 13115 Saint-Paul-lez-Durance, France; 2Univ. Rennes, Inserm, EHESP, Irset (Institut de Recherche en Santé, Environnement et Travail), UMR_S 1085, 35000 Rennes, France; 3Institut de Radioprotection et de Sûreté Nucléaire (IRSN), Pôle Santé Environnement-Santé (PSE-Santé)/Service de Recherche en Dosimétrie (SDOS)/Laboratoire de Micro-Irradiation, de Métrologie et de Dosimétrie des Neutrons (LMDN), Cadarache, 13115 Saint-Paul-lez-Durance, France; 4MARBEC, Univ. Montpellier, CNRS, Ifremer, IRD, INRAE, 34250 Palavas Les Flots, France

**Keywords:** gamma irradiation, teleost, anxiety-like behavior, sociability, neurotransmitter system, neurohormone

## Abstract

**Simple Summary:**

The increasing use of radiopharmaceuticals for medical diagnostics and radiotherapy raises concerns regarding health risks for both humans and the environment. Additionally, in the context of major nuclear accidents like in Chernobyl and Fukushima, very little is known about the effects of chronic exposure to low and moderate dose rates of ionizing radiation (IR). Many studies demonstrated the sensibility of the developmental brain, but little data exists for IR at low dose rates and their impact on adults. In this study, we characterized the molecular mechanisms that orchestrate stress behavior caused by chronic exposure to low to moderate dose rates of IR using the adult zebrafish model. We observed the establishment of a congruent stress response at both the molecular and individual levels.

**Abstract:**

High levels of ionizing radiation (IR) are known to induce neurogenesis defects with harmful consequences on brain morphogenesis and cognitive functions, but the effects of chronic low to moderate dose rates of IR remain largely unknown. In this study, we aim at defining the main molecular pathways impacted by IR and how these effects can translate to higher organizational levels such as behavior. Adult zebrafish were exposed to gamma radiation for 36 days at 0.05 mGy/h, 0.5 mGy/h and 5 mGy/h. RNA sequencing was performed on the telencephalon and completed by RNA in situ hybridization that confirmed the upregulation of oxytocin and cone rod homeobox in the parvocellular preoptic nucleus. A dose rate-dependent increase in differentially expressed genes (DEG) was observed with 27 DEG at 0.05 mGy/h, 200 DEG at 0.5 mGy/h and 530 DEG at 5 mGy/h. Genes involved in neurotransmission, neurohormones and hypothalamic-pituitary-interrenal axis functions were specifically affected, strongly suggesting their involvement in the stress response behavior observed after exposure to dose rates superior or equal to 0.5 mGy/h. At the individual scale, hypolocomotion, increased freezing and social stress were detected. Together, these data highlight the intricate interaction between neurohormones (and particularly oxytocin), neurotransmission and neurogenesis in response to chronic exposure to IR and the establishment of anxiety-like behavior.

## 1. Introduction

Ionizing radiation (IR) causes oxidative stress and genotoxic damages that can interfere with cellular activity and, directly or indirectly, with biological functions. Among their deleterious effects, cancer was originally found to be one of the main risks associated with exposure to IR [1,2,3]. In addition, epidemiological data on Japanese atomic bomb survivors (Hiroshima and Nagasaki in 1945) also demonstrated an elevated risk of developmental neurological disorders, especially microcephaly, in the prenatally exposed population [4]. Similar observations were taken in wild species exposed to IR due to nuclear power plant (NPP) accidents. After Chernobyl (1986) or Fukushima Daiichi (2011), yearling birds and *macaca fuscata* fetuses from contaminated areas displayed decreased brain size [5,6]. However, the majority of the studies described in the literature covers high and acute dose rates of IR (>6 mGy/h, [7]), and the biological effects of low dose rate radiation remains a subject of broad societal and scientific questioning and concern [8]. The impact of chronic exposure to low to moderate dose rates of IR on adult brain functions remains largely unknown and their impact on behavior and reproduction or survival is currently little studied [9].

Brain development is particularly sensitive to IR and other stressors (infections, ischemia, toxicants) [10,11]. Prenatal exposure in rodents to high doses of IR (more than 1 Gy) or X-rays reduces the number of neurons [12,13], decreases the complexity of dendritic arborization [14] as well as the number of neuronal progenitors after adults’ exposure [15] and can finally result into behavioral and memory defects [16,17]. In addition, recent work in humans demonstrates that IR can also alter brain homeostasis and function. Indeed, prenatal exposure to IR was suggested to impact cognitive functions with reduced school test results, lower verbal IQ or lower logical reasoning in adolescents [18,19,20]. However, the impact of IR is not restricted to the development period and was also proposed in adults with a decline in the accuracy and efficiency of cognitive performance [21,22]. Moreover, the effects of IR on adult neural stem cells or proliferating neural precursor cells were also demonstrated in the subventricular zone and the hippocampus of rodents [23,24,25,26,27,28]. In the context of chronic exposure to low dose rates of IR, fewer data exist on the effects on the central nervous system. For instance, an elevated risk of Parkinson’s disease was observed in a cohort of nuclear Russian workers which was strongly suggested to be linked with chronic exposure to external γ-rays and neutrons [29]. Moreover, studies in mice demonstrated that exposure to a much lower dose (0.1 Gy) could lead to the downregulation of neurotransmitters and genes with functions in synapses and neuron projections [30,31]. These results support the notion that the cellular response to chronic low dose and high dose and dose rates are different and further describe the emerging evidence of the neurological effects of exposure to radiation at dose rates much lower than previously thought in humans [29,32,33,34].

Stress response and anxiety-like behaviors are commonly quantified to assess the effects of toxicants on brain function [35,36]. In response to environmental stress, the adrenocorticotropic hormone (ACTH) is released by the pituitary gland and acts on the adrenal gland to stimulate the synthesis and secretion of glucocorticoids, which will in turn orchestrate the stress response via cortisol release [37]. This neuroendocrine system, denominated as the hypothalamic–pituitary–adrenal (HPA) axis, is conserved in mammals and fish (given that the interrenal organ in teleost is the homologue of the adrenal gland) and controls the stress response. The peripheral stress response induced by ACTH is controlled centrally by the release of the corticotropin release factor (CRF). In addition, other hypothalamic neurohormones such as oxytocin or arginine vasopressin also play a role in the regulation of the stress response. Indeed, the ablation of oxytocin expression in knockout mice induced anxiety-like behavior and increased the corticosterone level in the blood after the psychogenic stressor [38]. Additionally, stress elicited by forced swimming induced oxytocin release in the hypothalamus and peripheral circulation in rodents [39], which thus demonstrated an intricate interaction between oxytocin and the stress system. Moreover, in rodents, oxytocin can interact with the dopaminergic system or the serotoninergic system to influence a social response and to regulate anxiolytic effects, especially via the 5-HT1A and 5-HT2A/C receptors [40,41,42,43,44,45,46,47]. In zebrafish, somas of oxytocin neurons are found in the parvocellular preoptic nucleus corresponding to the paraventricular nucleus of the hypothalamus in rodents and in the periventricular nucleus of the posterior tuberculum [48,49,50]. As in mammals, oxytocin was also demonstrated to play a role in sociability in the zebrafish, notably in the development and maintenance of social behavior through oxytocin receptors (oxtr and oxtrl) [49,51]. Consequently, modifications in behavior could impact foraging and predator avoidance and interfere with reproduction.

With its simple cortical organization and an abundant constitutive neurogenesis, the zebrafish model, *Danio rerio*, is of great interest to study molecular mechanisms and behavior [52,53]. Moreover, this model accommodates easily to laboratory conditions and demonstrates a high degree of sociability thus making zebrafish a robust model for neuroscience research.

In this study, we aimed at characterizing the effects of IR impact on stress and behavior using adult zebrafish as a model system. The fish were exposed to dose rates of 0.05 mGy/h, 0.5 mGy/h and 5 mGy/h for 36 days. Those exposure levels correspond to the domain of the low dose rates (<6 mGy/h, [7]), are near the reference value for ecosystem radioprotection (0.010 mGy/h) [54] and are in the range of the dose rates estimated after the Chernobyl accident (0.4 mGy/h) [55]. A multiscale approach was used to evaluate the effects on adult behavior, with emphasis on sociability, stress and locomotion. The molecular effects of chronic exposure to IR on the brain were assessed through a transcriptomic approach, completed by the study of spatial changes in gene expression of *oxytocin* and *cone rod homeobox* by RNA in situ hybridization.

## 2. Materials and Methods

### 2.1. Zebrafish Maintenance and Irradiation

Adult zebrafish were maintained at the Institut de Radioprotection et de Sûreté Nucléaire facilities in accordance with French and European regulations on the protection of animals used for scientific purposes (EC Directive 2010/63/EU and French Decret 2013–118). All experiments were approved by the ethical comity (C2EA, IRSN, Fontenay-aux-Roses, France) and the French Ministry of Research under the reference APAFIS#11488. Six- to nine-month-old wildtype zebrafish of the AB strain were provided by Amagen (Gif-sur-Yvette, France) and Sorbonne Paris University (Institut de Biologie Paris Seine, Paris, France). Constant conditions were preserved for fish maintenance: 27  ±  1 °C, 350–450 µS/cm, pH 7.5 and 12/12 h dark-light cycles, in the ZebTEC system (Techniplast, Décines-Charpieu, France). The fish were fed three times a day with Gemma Micro 600. Any mortality was noted, the homogeneity of the fish’s weight and length were verified after behavior experiments, and no difference was observed in any condition or between males and females (data not shown). The fish were chronically exposed for 36 days to gamma irradiation by a ^137^Cs source in the MICADO experimental irradiation facility (IRSN, Cadarache, Saint-Paul-lez-Durance, France), except for a duration of one hour five times a week without irradiation to visually verify the fish’s health. Three dose rates were employed: 0.05 mGy/h, 0.5 mGy/h and 5 mGy/h. To confirm the dose rates of exposure, we used operational dosimetry with radio-photoluminescent dosimeters (RPL, GD-301 type, Chiyoda Technol Corporation Japan, Tokyo, Japan), and the effective dose rates were estimated by MCNPX v2.70 (Appendix A).

### 2.2. Behavioral Analysis

#### 2.2.1. Novel Tank and Social Preference Tests

The Novel Tank Test (NTT) was used for the assessment of the stress behavior and locomotor profiles of adult zebrafish. Single fish (*n* = 20 females and males per condition) were placed for 6 min in a tank (27.9 × 7.1 × 15.2 cm) filled with water (13.5 cm deep). Videos were recorded from the side using a Basler acA1300–60 gm camera fitted with an infrared filter detection (Infrarot 850, Heliopan, Noldus Information Technology, Wageningen, Holland) and a bespoke infrared spotlight platform to avoid shadows and artefacts. To study the locomotion, we measured the total distance travelled. For behavioral parameters related to stress, we designed an arena in Ethovision XT14 that separated the tank into top and bottom zones of equal dimensions and measured the following parameters: time spent and number of entries in top zone, latency to first entry to the top zone and immobility time. Mobility was defined as the degree of the displacement of the zebrafish’s body independent of spatial movement of the center body point [56].

The social preference test was performed as previously described in order to assess how the fish interacts towards a social stimuli [57,58]. One minute after the end of the NTT, this trial was performed for a total of 5 min using the same tank as for the NTT to avoid stress, except that a new compartment of 15 × 7.1 × 15.2 cm with four stimulus fish (two females, two males) was located on one side of the central arena. We divided the central arena that includes the test fish into two equal parts (non-social and social zones). An additional zone measuring half the body length of the fish (Z1) was included in the social zone close to stimulus fish (Figure 2. The size of that zone was chosen so the fish swimming in Z1 presented a social and non-aggressive behavior. Locomotion and stress were measured by the following parameters: total distance travelled and immobility time. Sociability parameters were assessed by measuring: the time spent and number of entries in the social zone and time spent in Z1 (Figure 2). For both assays, the zebrafish were transported to the behavioral testing room the evening before testing and acclimated for 15 h. Ethovision 14XT (Noldus) was used for tracking and analysis.

#### 2.2.2. Shoaling Test

Group social behavior was analyzed by using the shoaling test [59] in an opaque plastic tank of 61 cm diameter filled with 5 cm of water depth. The zebrafish were transported to the behavioral testing room the evening before testing and acclimated for 15 h. Groups of eight familiar fish (four females, four males) were placed at the same time in the middle of the tank, left to acclimate for 5 min and filmed from above for 10 min with the same setup as the test used for NTT. This experimental setting was modified from Pham and colleagues and Gutiérrez and colleagues [48,60]. The analysis was formed on four different shoals per condition. Ethovision 16XT (Noldus) was used for tracking and measuring locomotion (total distance travelled), stress (immobility time) and inter-individual distances. A custom code for EthoVision XT 16 written in JavaScript, kindly provided by Fabrizio Grieco and Egon Schrasser, was employed for the nearest neighbor distances.

### 2.3. Acetylcholinesterase Activity

To avoid any neuronal modification that could be caused by anesthetic, the fish were euthanized by hypothermic shock on ice (2–4 °C) in agreement with the ethical comity and any opposition from the French Ministry of Research under the reference APAFIS#11488. Their heads were sectioned in the frontal plane and the brain and muscles were dissected under binoculars. The organs (*n* = 15–17 females and males per condition) were homogenized in a homogenization buffer (Phosphate buffer, Glycerol 20% *v*/*v*, PMSF 0.2 mM) using a bead crusher (Precellys Evolution, Bertin Technologies, Montigny-le-Bretonneux, France) maintained at 4 °C during homogenization, at 5000 rpm and for two cycles of 20 sec. We used the Bradford method for protein assay to determine the total protein concentration in each sample. The measure of acetylcholinesterase (AChE) activity was performed in 96-well plates as follows. We added 200 µL of Phosphate buffer, 4 µg of protein sample in duplicates or AChE standard and 20 µL of dithiobisnitrobenzoate (0.2 mM final concentration) (DTNB D218200, Sigma, Setagaya City, Japan) before incubation for 10 min at 25 °C. Then, 10 µL of Acetylthiocholine iodide (2 mM final concentration) (ATCi A5751, Sigma) were added. The hydrolysis of acetylthiocholine was monitored by the formation of the thiolate dianion of DTNB, and the absorbance was read at 412 nm for 3 min at 30 s intervals, as previously described [61]. The AChE activity was expressed as U/mg defined by the number of enzyme units (amount of enzyme that catalyzes the reaction of 1 nmol of substrate per minute) per ml divided by the concentration of protein in mg/mL.

### 2.4. Library Preparation and RNA Sequencing

The fish were euthanized by hypothermic shock on ice (2–4 °C), their heads sectioned in the frontal plane, and the brain was dissected under binoculars. Each sample consisted of three to four dissected telencephalons formed in four to six replicates, keeping an equilibrated number of males and females per condition. The brains were homogenized in 200 µL of Trizol (Life Technologies, Carlsbad, CA, USA) using beads homogenizer (Precellys Evolution, Bertin Technologies) with two cycles of 30 s each. Total RNA extraction was performed following manufacturer instructions with an additional chloroform extraction before isopropanol precipitation. RNA integrity (RIN), quality and concentration were assessed using RNA Nano Chips (Bioanalyzer 2100, Agilent, Santa Clara, CA, USA). All samples had a RIN > 8. Sequencing libraries were generated from 1 µg of the total RNA following the TruSeq mRNA stranded protocol (Illumina, San Diego, CA, USA). After a quality check and concentration determination on DNA1000 Chips (Bioanalyzer 2100, Agilent, Les Ulis, France), the libraries were run on a HiSeq4000 platform to produce 50 bases long paired-end reads (GenomeEast, IGBMC, Illkirch, France).

### 2.5. Analysis of Transcriptomics Data

Between 72 and 136 Million of good-quality reads (Q > 30) were produced for each sample and analyzed as described before [62]. Briefly, the reads quality was assessed with an in-house pipeline to determine quality over read length, duplication rates, insert size, adaptors contamination, nucleotide composition and mapping rates against the reference genome (*RNA-STAR* with GRCz11 and known exon–exon junctions from Ensembl release 98). Normalization and differential expression analysis were performed with *DESeq2 v1.30.1*. Biological reproducibility was assessed by the hierarchical clustering of the variance stabilized expression data (rlog) obtained from *DESeq2* with Pearson’s correlation and complete linkage method. Genes with |fold change| ≥ 1.5 and *p*-value corrected for multiple testing, fdr < 0.05 (false discovery rate [63]) were considered as differentially expressed in all analysis. Afterwards, functional enrichments was performed using Gene ontology (GO) with R package *clusterProfiler* using zebrafish genes and human orthologues retrieved by Ensembl biomart (as published before, [64,65,66]) as gene annotation is more abundant in this species. The significance was considered when fdr from Fisher’s exact test was <0.05. For the volcano-plot of differentially expressed genes (DEG) of neurohormones and neurotransmitters, the following GO terms were used to select genes with the R package *biomaRt* [64] (Appendix A). The resulting list of genes was then manually curated to verify gene function.

### 2.6. RNA In Situ Hybridization on Adult Brain Sections

Antisense RNA probes labelled with digoxigenin (DIG) for *oxt* and *crx* were generated from adult zebrafish brains as follows. Total RNA extraction was performed using TRIzol/chloroform extraction (Life Technologies), with a supplementary chloroform extraction step before performing the isopropyl alcohol precipitation of the nucleic acids. We performed the reverse transcription using SuperScript III (Life Technologies), dNTP, polydT primers and 1 µg of total RNA following manufacturer instructions. PCR amplification was performed using Taq platinum kit (Life Technologies) with 1 µL of RT and the primers pairs: *oxt* forward AATGTCTGGAGGTCTGCTGT, *oxt* reverse TGCACTAATGTACAGTCAAGCT, *crx* forward GCTGCTCGGTCTCTTATTGC, *crx* reverse AGTCAAGGCAGTCTACCGAA. PCR products were validated by electrophoresis on 1.2% *w*/*v* agarose gel and cloned into a PCRII dual promoter bacterial vector following the kit instructions (TOPO TA Cloning, Invitrogen, Waltham, MA, USA). The transformation of chemically competent TOP10′E.coli (Invitrogen) was performed with 2 µL of ligation production. The bacteria were plated on a Petri dish with LB agar with 100 µg/mL Ampicillin and incubated overnight at 37 °C. Five colonies for each construct were picked and grown individually in liquid LB medium with 100 µg/mL of Ampicillin overnight at 37 °C with agitation. MiniPreps (Qiagen DNA MiniPreps, Hilden, Germany) were performed for the extraction of plasmid DNA following the alkaline lysis protocol and sent for Sanger sequencing to check cDNA identity and the orientation of cloning in the PCRII vector. The digestion of the DNA was performed by cutting the DNA plasmid at the 5′ end of the cDNA with restriction enzyme: NotI HF, followed by phenol/chloroform extraction of the linearized vector, and in vitro transcription of the Dig RNA probes with Sp6 RNA polymerase and the Dig-UTP labelling mix (Roche, Basel, Switzerland). The technique of in situ hybridization labelling was adapted from ZFin [67]. Two–three brains were used per condition. Briefly, freshly dissected brains were fixed overnight at 4 °C in 4% *w*/*v* PAF and transferred in 100% *v*/*v* methanol at −20 °C until use. The brains were rehydrated in a decreasing gradient series of methanol diluted in PBS *v*/*v* at 70%, 50% and 25% for 5 min each and then washed four times 5 min in PBS with 0.01% *v*/*v* Tween 20 (PTW buffer) and incubated with proteinase K at 10 µg/mL in PTW buffer for 30 min and fixed in 4% *w*/*v* PAF for 20 min. Five consecutive washes of 5 min in PTW were completed before incubation with hybridization buffer (50% *v*/*v* deionized formamide, 5X SSC, 500 µg/mL yeast tRNA, 50 µg/mL heparine, 0.1% *v*/*v* Tween20, 9 mM citric acid in water) that lasted until all the brains reached the bottom of the tube (approximately 20 min). The brains were then incubated 4 h at 65 °C in clean RNA hybridization buffer and put in contact overnight with 50 ng (*oxytocin*) or 100 ng (*crx*) of gene specific DIG-labelled RNA probes diluted in RNA hybridization buffer. The brains were washed at 65 °C as follows: twice during 30 min in wash buffer 1 (50% *v*/*v* formamide, 1X SSC, 0.05% *v*/*v* Tween20), once in wash buffer 2 (2X SSC, 0.1% *v*/*v* Tween20) for 15 min, twice in wash buffer 3 (0.2X SSC, 0.1% *v*/*v* Tween20) for 30 min and finally once in wash buffer 4 (0.1X SSC, 0.1% Tween20) for 5 min. The brains were then washed 3 times 5 min in PTW at room temperature, incubated 1 h at room temperature in blocking buffer (1X PBS, 0.1% *v*/*v* Tween20, 0.2% *v*/*v* BSA, 1% *v*/*v* DMSO), embedded into 2% agarose and sectioned at 70 µm with a vibratome (VT1200S Leica Biosystems, Wetzlar, Germany) in a coronal plane. The brain sections were incubated 1h in blocking buffer, then overnight at 4 °C with anti-Digoxigenin-AP Fab fragment antibody (Roche Diagnostics GmbH, Mannheim, Germany) diluted at 1:4000 in blocking buffer and washed 5 times 15 min with PTW buffer before staining with NBT/BCIP (Roche Diagnostics GmbH, Germany). The slides were mounted with Vectashield HardSet (Vector Laboratories, Newark, CA, USA, ref: H-1400). Each in situ hybridization staining was repeated at least two times. The expression patterns were observed under a Leica binocular (DM750) equipped with a Leica ICC50 camera at ×10, ×20 and ×40 zoom. For the quantification of the intensity signal of oxytocin, we performed a fluorescent RNA in situ hybridization. The protocol is similar as the one previously described except that before adding the hybridization buffer, the samples were incubated for one hour with 3% hydrogen peroxide (H_2_O_2_, Sigma) in PTW and then washed three times 10 min in PTW; anti-digoxigenin-AP was substituted by Anti-digoxigenin-POD (poly) Fab fragments (Roche Diagnostics Gmbh, Germany) diluted at 1:1000 in blocking buffer and before staining with TSA Plus Cyanine 3 (Perkin Elmer, Boston, MA, USA), samples were rinsed twice 10 min in 0.002% H2O2 in PTW. Zeiss Confocal LSM780 was used with laser 561 at ×40 zoom. We first checked for autofluorescence and background noise without detecting either of them.

### 2.7. Statistical Analysis

For the behavioral analysis, and especially novel tank and social preference tests, a first statistical analysis by a two-way ANOVA test indicated significant sex differences (data not shown). This is in agreement with other studies that already showed distinct behavioral profiles between males and females [68,69]. We thus formed analyses for each sex separately. Linear effects models with Gaussian family were used for normally distributed data (total distance travelled in NTT, latency to first entry to the top zone, time in Z1, time of immobility) or with sqrt transformation to reach normality when needed (time in top zone, total distance travelled for the social preference test). For over-dispersed counted data, the generalized linear mixed effects model for the negative binomial family was used (number of entries in top or social zone). As the experiments were performed in two different batches, we added a random effect corresponding to the session date. *p*-values < 0.05 were considered significant.

Regarding the Shoaling test, statistical analysis was formed by applying a generalized linear mixed model with Gamma distribution for continuous and positive data. Since individuals show more behavioral similarities among one shoal, the shoals were used as a random effect in the model. *p*-values < 0.05 were considered significant.

The data corresponding to acetylcholinesterase activity analysis being normally distributed (except for female muscles that needed log transformation), the analysis was performed on each sex separately by one-way ANOVA followed by a Dunnet post-hoc test. *p* values < 0.05 were considered significant.

For the quantification of fluorescent RNA in situ hybridization of *oxytocin*, equal regions of interest (ROIs) were selected on several vibratome slices of three brains per condition and the integrated density for each ROI was determined by dividing the intensity of ROI to a representative area without signal (the removal of background noise). Statistical analysis was undertaken by applying a generalized linear mixed model with Gamma distribution for continuous and positive data. Since several slides originated from the same brain, brains were used as a random effect in the model. *p*-values < 0.05 were considered significant.

## 3. Results

### 3.1. Chronic Irradiation Altered mRNA Expression in Adult Zebrafish Telencephalon

The modulation of transcriptional expression in adult telencephalons was assessed by RNAseq after continuous exposure to gamma radiation for 36 days at 0.05 mGy/h (D005), 0.5 mGy/h (D05) and 5 mGy/h (D5). A dose rate-dependent increase in differentially expressed genes (DEG) was observed with 27 DEG at 0.05 mGy/h, 200 DEG at 0.5 mGy/h and 530 DEG at 5 mGy/h (|fold change| ≥ 1.5 and fdr < 0.05). A relatively high number of DEG was common between 0.5 mGy/h and 5 mGy/h (*n* = 156 genes, 26%) while very few DEG were shared with the 0.05 mGy/h condition (Figure 1a).

The enrichment of Gene Ontology (GO) Molecular Function and Biological Process, either with *Danio rerio* genes or via human orthologous genes, demonstrated that the pathways altered after chronic exposure to IR at D05 and D5 included visual perception (GO:0007601; D05: fdr < 10^−22^ D5: fdr < 10^−17^), the regulation of G protein-coupled receptor signaling pathways (GO:0008277; D05: fdr < 10^−8^; D5: fdr < 10^−8^) and serotonin metabolic process (GO:0042428; D05: fdr = 0.002; D5: fdr = 0.03) (Figure 1b and Appendix A). We also detected other biological processes that were significantly enriched such as retinoid binding (GO:0005501; D05: fdr = 10^−5^; D5: fdr = 10^−4^), the regulation of axon extension involved in axon guidance (GO:0048841, D05: fdr = 0.01; D5: fdr = 0.03) and optokinetic behavior (GO:0007634; D05: fdr = 0.01; D5: fdr = 0.03) (Appendix A). In contrast, pathways enriched after irradiation at 0.05 mGy/h highlighted the dysregulation of genes involved in protein refolding (GO:0042026; fdr < 10^−4^).

### 3.2. Chronic Irradiation Induced Hypolocomotion

Since transcriptomics data revealed changes in gene expression in the brain after irradiation, we assessed if these modifications could result in potential changes in behavior. Since DEGs were mostly observed when exposing at 0.5 mGy/h and 5 mGy/h, we focused on these two dose rates as modifications after 0.05 mGy/h radiation perturbed the expression of few genes (*n* = 27). First, locomotion was measured as an integrated parameter to evaluate the effects of radiation on adult brain function and behavior. The total distance travelled was measured by three different behavioral tests on adult zebrafish, the shoaling test (ShT), the Novel Tank test (NTT) and the social preference test (SP) (Figure 2).

The total distance travelled was decreased at both dose rates in the ShT (D05: t = −13, *p* = 10^−16^; D5: t = −4.7, *p* = 10^−6^, Figure 3a). Similar trends were also detected in the females exposed in NTT at 0.5 mGy/h (t = −2.7, *p* = 0.01) and 5 mGy/h ( t = −1.4, *p* = 0.17) and in the SP (D05: t = −1.5, *p* = 0.13; D5: t = −2.7, *p* = 0.01), but not in males (Figure 3b–e). Since acetylcholine is involved in muscle contraction and in brain function by participating in anxiety-like behavior or social response [70,71], the activity of acetylcholinesterase (AChE) was measured in both the muscle and brain after irradiation. Decreased reduction in enzymatic activity can be observed in the brain and muscle of exposed females at both dose rates but this decrease is not statistically significant (in brain: F = 2.5, *p* = 0.11; in muscles: F = 2.77, *p* = 0.09, Figure 3f,g). We did not observe any effect in the brain or muscles in males (brain: F = 0.15, *p* = 0.86; muscles: F = 0.78, *p* = 0.47, Figure 3h,i).

### 3.3. Chronic Irradiation Increases Stress Behavior in Female Zebrafish

We then evaluated the stress potentially induced by IR by measuring the number of entries, the latency to first entry and the time spent in top zone in the NTT. The number of entries in the top zone reached statistical significance only for females exposed at 0.5 mGy/h (Figure 4, t = −3.1, *p* = 0.002).

The other parameters measured in the NTT (latency to first entry and cumulative duration in the top zone) were not altered (Figure 4a,b), middle and right panel). Immobility is often used to evaluate anxiety-like behavior in zebrafish [72]. We thus assessed the relative time of immobility (freezing) in the three behavioral tests as another way to evaluate stress after chronic exposure to radiation. An increase in immobility was observed in females but not in males at both dose rates in the SP (D05: t = 2.3, *p* = 0.03; D5: t = 2.4, *p* = 0.03, Figure 4d). A similar increase was also observed in the NTT, especially at 0.5 mGy/h (D05: t = 2.7, *p* = 0.01; D5: t = 1.4, *p* = 0.17, Figure 4c), as well as in the ShT but this alteration was not statistically significant (D05: t = −1.4, *p* = 0.17; D5: t = −0.54 *p* = 0.59, Figure 4e).

### 3.4. Chronic Irradiation Reduces Sociability in Female Zebrafish

The zebrafish being a social animal with complex behavior, the proximity of fish with conspecifics was measured to assess sociability in the SP test. We observed that the time spent close to the conspecifics (Z1) was decreased for females in both irradiated conditions but only significant in 0.5 mGy/h of the irradiated fish (D05: t = −2.7, *p* = 0.01 and D5: t = −1.9, *p* = 0.06, Figure 5a). We also measured the time spent and number of entries with a one-body-length zone close to the conspecifics and observed a significant decrease in entries in 5 mGy/h of the irradiated fish (data not shown).

A significant decrease in the number of entry in the social zone was also detected using a broader zone at 5 mGy/h (t = −2.4, *p* = 0.02, data not shown). We did not observe any modifications in the males. We then used the ShT to assess group cohesion within a shoal. No alterations in the social group was observed using either nearest neighbor or inter-individual distances (Figure 5b,c).

### 3.5. Induction of Gene Expression Involved in Stress Response or Neurogenesis

Since the zebrafish displayed stress behavior and the transcriptomic profile of the brain radiation response suggested the dysregulation of genes involved in neurotransmission (including serotonin), we explored the changes of neurohormones and neurotransmitters expressions in the RNAseq dataset (Appendix A. The significant modulation of eight genes coding for proteins with a function in neurohormone synthesis, uptake, metabolism and receptor production, or in neurotransmission was found at 0.5 mGy/h and thirty-two genes at 5 mGy/h (Figure 6a,b and Appendix A).

These included the genes coding for the enzymes *tryptophan hydroxylase 1a* (*tph1a*) and *tryptophan hydroxylase 2* (*tph2*) that control brain serotonin synthesis, as well as the neurohormones *oxytocin* (*oxt*) and *arginine vasopressin* (*avp*) (Figure 6c). We also detected a variation in the expression of *cyp11c1* involved in cortisol and *11-ketotestosterone* (*11-KT*) production, *asip2b* known to regulate cortisol secretion and the stress-induced transcription factor *nr4a1* [73,74,75,76]. To further explore how radiation can impact gene expression in the brain, and validate the RNAseq data, we performed RNA in situ hybridization on adult brain sections for *oxt*, a gene expressed in the preoptic area and involved in both the regulation of stress and social behavior [77]. At 5 mGy/h, we observed an ectopic expression of *oxt* in the ventricular zone (VZ) of the anterior part of the parvocellular preoptic nucleus (PPa)(Figure 7a,a’) where neural progenitors are located [78]. Indeed, in the control condition, cells lining the VZ do not express *oxt*, while after irradiation, *oxt* expression is detected. To quantitatively confirm this observation, we measured *oxt* expression in control and irradiation (5 mGy/h) by fluorescent RNA in situ hybridization and detected a significant increase in signal intensity (Appendix A).

Similarly, the gene *cone-rod homeobox* (*crx*), known to be involved in neuronal differentiation in the retina [76], was strongly upregulated in the PPa in both the VZ as well as in cells located laterally to the VZ (Figure 7b,b′). These results confirm the upregulation of *oxt* and *crx* observed at 5mGy/h by the transcriptomic analysis.

## 4. Discussion

In this study, we used adult zebrafish to better characterize the effects of chronic low to moderate dose rates of IR on brain function and behavior. We employed a multiscale approach that characterized the effects on behavior, including locomotion, the assessment of the stress response and sociability, completed by a transcriptomic analysis on the telencephalon to characterize the effects at the molecular scale. This analysis was completed by a detailed mapping of the gene expression of two candidate genes using RNA in situ hybridization on adult brain sections. This strategy allowed us to assess which molecular pathways were dysregulated following chronic exposure to low and moderate dose rates of IR and helped in deciphering the molecular basis of changes in adult behavior.

Transcriptomic analysis highlighted the dysregulation of genes with function in synaptic plasticity, including axon guidance signaling (*arl3l2*, *arl3l*, *sema3d* or *unc119.2*) or the regulation of the G protein-coupled receptor signaling pathway (*cng1*, *pde6b*, *grk7*, *guca1c*). It is well known that synaptic plasticity is involved in organisms’ adaptation and, when disturbed, may lead to pathological behaviors such as stress-related disorders [79,80]. A previous study in mice exposed acutely to 100 mGy detected similar perturbation, but in contrast to our results, gene expression was mostly downregulated in the rodents [30]. Additionally, high doses of acute exposure includes the induction of neuro-inflammation with the overexpression of pro-inflammatory genes [81,82], cellular senescence [83], the decrease in adult neurogenesis possibly due to the apoptosis of proliferating cells [82,84], DNA damage like double strand breaks [10,85] and oxidative stress through the elevation of pro-oxidants or lipid peroxidation [86,87]. Interestingly, in our study, we could not detect significant change in these processes through the transcriptomic analysis. Possibly, the different exposure scenarios (chronic versus acute) or the use of young 8- to 10- week mice (while we used sexually mature organisms) explain this difference, but more likely, given the very high regenerative capacity of the adult zebrafish brain in comparison to juvenile rodent [88,89,90], the genes involved in the axon guidance and G protein-coupled receptor signalling were only upregulated in fish, which might reflect a specific response to the insults caused by IR. This hypothesis of induced regenerative process after irradiation is supported by the very strong upregulation of *crx* in the adult telencephalon. Indeed, this gene is a well-known transcription factor (TF) of the otx-like homeodomain family that regulates the neuronal differentiation of photoreceptors in the retina [91]. However, its role in the adult zebrafish telencephalon is currently unknown. Nevertheless, its expression in cells lining the ventricular zone (VZ) as well as in cells located more laterally in the PPa corresponds to the expression domain of neuronal progenitors (lining the VZ) and differentiating neurons (laterally to the VZ). The fact that *crx* is strongly upregulated in the PPa after irradiation could thus indicate the production of new neurons in this region of the adult brain and highlights its potential role in neurogenesis. If these transcriptomic modifications lead to changes at the protein level remains to be investigated. More analysis, including double labelling with progenitors and differentiation markers, is needed in order to confirm or infirm this regenerative hypothesis.

We observed a hypolocomotion state noticeable by a reduced total distance travelled at dose rates equal to and higher than 0.5 mGy/h in the ShT, which was also detectable in the SP and the NTT. Interestingly, similar effects on locomotion were identified in 5-day-old larvae exposed to continuous radiation at the same dose rates [62]. Moreover, hypolocomotion was observed in mice 6 h after acute exposure to 500 mGy or 2000 mGy [92], and up to 18 months after irradiation of 10-week-old mice at 500 mGy [93]. Thus, our data complement the existing knowledge of IR by showing the effects on locomotion in adults in the field of low dose rates. Our analysis on AChE activity, the enzyme involved in the degradation and recycling of acetylcholine that acts both in the brain and neuromuscular junctions, demonstrated a marginal decrease in enzymatic activity in the female muscles and brain at 0.5 mGy/h and 5 mGy/h. In males, a non-significant increase in AChE activity was observed in the muscles at 5 mGy/h, and we did not see any effect on locomotion or stress. Together, our data highlight chronic irradiation-induced hypolocomotion in females, and this effect was substantially independent of possible alterations at the neuromuscular junction. This result contrasts with previous data that showed decreased AChE activity in zebrafish larvae exposed at 0.01 mGy/h [94].

The results on stress behavior from the NTT revealed a decreased number of entries in the top zone after the exposure of females at 0.5 mGy/h, suggestive of increased stress, but no change could be observed at 5 mGy/h, and males were not affected. If a non-linear dose rate effect could be proposed for females, it remains possible that insufficient individual data impaired the analysis. The cumulative duration of immobility (also called freezing) is another parameter used to measure stress [95]. Our data demonstrated an increased cumulative time of freezing in females irradiated at 0.5 mGy/h in the NTT, but, as before for locomotion, no effect could be detected in males or at 5 mGy/h. However, by measuring this endpoint in the SP, we could detect a significant increase in freezing both at 0.5 mGy/h and 5 mGy/h in females. An increased cumulative duration of immobility was also apparent in the ShT, but these results were not statistically significant, presumably because males’ and females’ behavior cannot be distinguished in this group behavior test. The hypolocomotion is thus presumably a consequence of increased freezing time. These results show that chronic irradiation at 0.5 mGy/h, and presumably higher dose rates, induces stress-like behavior in female zebrafish but not in males. At the molecular level, stress behavior is linked to increased cortisol release. Under stress, an increase in CRH levels results in higher glucocorticoid hormone (GC) secretion (cortisol or corticosterone depending on species) which bind in the plasma to corticosteroid-binding globulin (CBG) and albumin [96]. As teleost have a developed hypothalamus-pituitary-interrenal axis and use cortisol as their primary stress hormone [73,97,98,99,100,101], the increasing freezing time observed in our data is very likely to be linked to an elevation of cortisol secretion. The fact that several genes involved in glucocorticoid production were upregulated in the brain of the irradiated zebrafish, including *cyp11c1*, *nr4a1*, *asip2b*, *avp* and *oxt*, is also in favor of such interpretation and furthermore can provide a mechanistic explanation for the increased stress behavior observed at dose rates equal to and higher than 0.5 mGy/h. These results are in agreement with previous data on the chronic exposure of meadow voles at 22.6 μGy/h [102], which demonstrated higher levels of free-corticosterone in females than in males, along with elevated corticosteroid-binding globulin (CBG) (in females). The results in meadow voles are thus suggestive of a chronic stress response visible by the prolonged production of GC and its transport in the plasma. The evaluation of such a response in the context of chronic irradiation remains to be determined.

The use of RNA in situ hybridization on adult brain demonstrated an upregulation of *oxt* in the anterior part of the parvocellular preoptic nucleus (PPa). It expresses several neurohormones such as oxytocin or arginine-vasopressin with conserved function compared to mammals [103] such as the regulation of water homeostasis [104], the control of blood pressure [105] as well as a role in social behavior and the regulation of anxiety [49,50,106]. Interestingly, a recent analysis of the stress response in zebrafish correlated to the increased number of oxytocinergic cells in the brain with environmental stress provoked by immersion in deionized water for 24 h [50]. Other experiments in rodents revealed that repeated defeat experiences increased the number of oxytocinergic neurons in the PVN as well as after chronic homotypic stress [107,108]. From these studies, it was proposed that acclimation to chronic stress might involve the upregulation of oxytocin expression in the hypothalamus. Therefore, chronic low to moderate dose rates of gamma radiation can induce the upregulation of *oxt* observed in the VZ of the PPa which could thus reflect the acclimation of fish to chronic stress.

The SP test revealed a decreased sociability of adult fish at both dose rates. No change in group behavior could be detected in the ShT, presumably because no distinction could be established between males and females in this test. Mechanistically, it is tempting to speculate that this change in social behavior is linked to changes in the expression of *oxytocin* we observed in the cells of PPa. Indeed, oxytocin has a dual role in anxiety-like behavior and in social behavior in both mammals [109,110,111] and zebrafish [49,51,112]. For instance, the knockout of one of the oxytocin receptors (*oxtr*−/− *or*
*oxtrl*−/−) enhanced the social preference level in zebrafish larvae at 3 and 4 weeks post fertilization [51]. Similarly, the early life ablation of oxytocin neurons led to a reduced social preference in adult zebrafish [113]. However, contrasting results in social behavior are obtained depending on the experimental design [49,51,114], which means it is more difficult to predict if an increase in oxytocin production will lead to increase or decrease in social preference. Beside oxytocin, we also noted the upregulation of *avp* expression in the brain after irradiation at 5 mGy/h (by RNAseq). This neuropeptide can stimulate ACTH secretion from the fish pituitary, and it was shown that after acute stress, such as confinement or enforced displacement, the *avt* transcript is induced in parvocellular AVT neurons in the rainbow trout [115,116]. Moreover, arginine vasopressin was also proposed to regulate social behavior since injecting avp in zebrafish induced social interaction [48]. Several studies suggested crosstalk between avp and oxt signaling in the modulation of social behavior [49,117,118], suggesting that both systems might act in synergy in the increased anxiety observed after irradiation. In zebrafish, oxytocin has been proven to not take part in the regulation of the stress response when acting through its receptors [49,119]. However, we cannot rule out that a decrease in the degree of sociability could be an indirect consequence of the stress response observed. Functional studies could be useful to precisely understand if these two responses (stress and sociability) are independent in the context of chronic low and moderate dose rates of ionizing radiation.

Finally, transcriptomic analysis also revealed the upregulation of *tph1a* and *tph2* at 0.5 mGy/h and 5 mGy/h, two genes coding for enzymes essential for serotonin synthesis. The serotonin levels in the brain were correlated to the maturation of shoaling behavior during zebrafish development [120]. Moreover, the interactions between the serotonin and oxytocin systems were suggested in humans and mice [46,47]. Serotonin plays also a role in the stress response by the activation of the HPA axis in mammals and studies in zebrafish with 5HT agonist/antagonist receptors showing an impact on anxiety-like behaviors [121,122]. Our results also showed the dysregulation of the transporter *slc18A2*, also known as *vesicular monoamine transporter 2* (or *VMAT2*), which transports either serotonin or dopamine. The knock out of the dopamine transporter (DAT) in zebrafish resulted in a chronic anxiety-like state [123]. Hence, the modification of serotonin and dopamine availability might participate in the behavioral patterns observed. The stress response observed after irradiation involved thus a complex interaction between various neurohormones and neurotransmission, which together drive the acclimation of fish subjected to chronic environmental stress.

## 5. Conclusions

In this study, we aimed at elucidating the relation between the molecular and behavioral effects under chronic low to moderate dose rates of IR. Our results show the emergence of a stress response, highlighted by genes known to be involved in the HPI axis or coding for neurohormones (oxytocin, arginine-vasopressin) and neurotransmitters (serotonin, dopamine but not acetylcholine) that play a role in the regulation of anxiety-like behavior. This stress response is translated at the individual scale by hypolocomotion, increased freezing and social stress. We demonstrated that the effects on locomotion were not linked to the alteration of AChE activity in the brain or at the level of neuromuscular junctions. Finally, we confirmed by RNA in situ hybridization the increased expression of *oxt* and *crx* in the VZ of the PPa, suggestive of modulations of neurohormone production in the PPa to regulate stress behavior. Together, these findings highlight the intricate interaction between neurohormones, neurotransmission and neurogenesis to balance disturbed brain homeostasis caused by chronic exposure to IR.

## Figures and Tables

**Figure 1 cancers-14-03793-f001:**
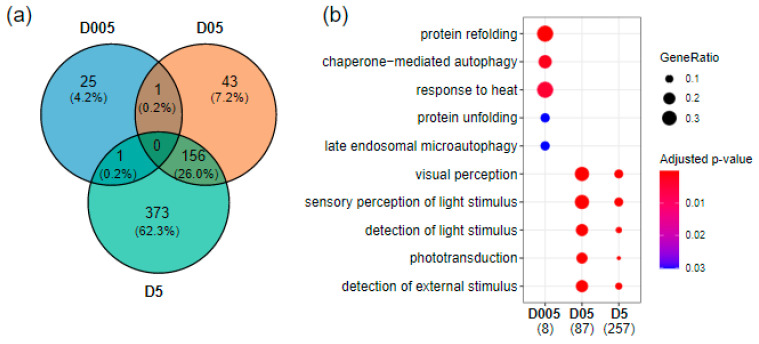
Transcriptomics effects of radiation exposure on zebrafish telencephalon. (**a**) Venn diagram of differentially expressed genes at the three dose rates (|fold change| ≥ 1.5 and adjusted *p*-value < 0.05. (**b**) GO enrichment of biological processes using human orthologues showing the top enriched pathways at D005 (0.05 mGy/h), D05 (0.5 mGy/h) and D5 (5 mGy/h). The numbers in brackets correspond to the number of genes present in the enriched GO terms. The dot size corresponds to the fraction of genes in a given GO term compared to all DEG (GeneRatio).

**Figure 2 cancers-14-03793-f002:**
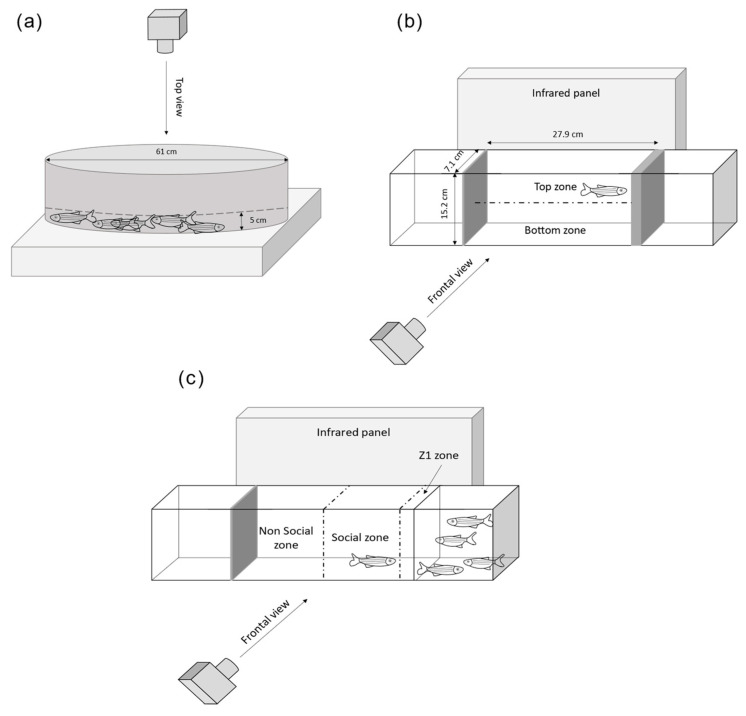
Experimental setup of the three behavioral tests. (**a**) The shoaling test, (**b**) the novel tank test, (**c**) the social preference test [57,58,59].

**Figure 3 cancers-14-03793-f003:**
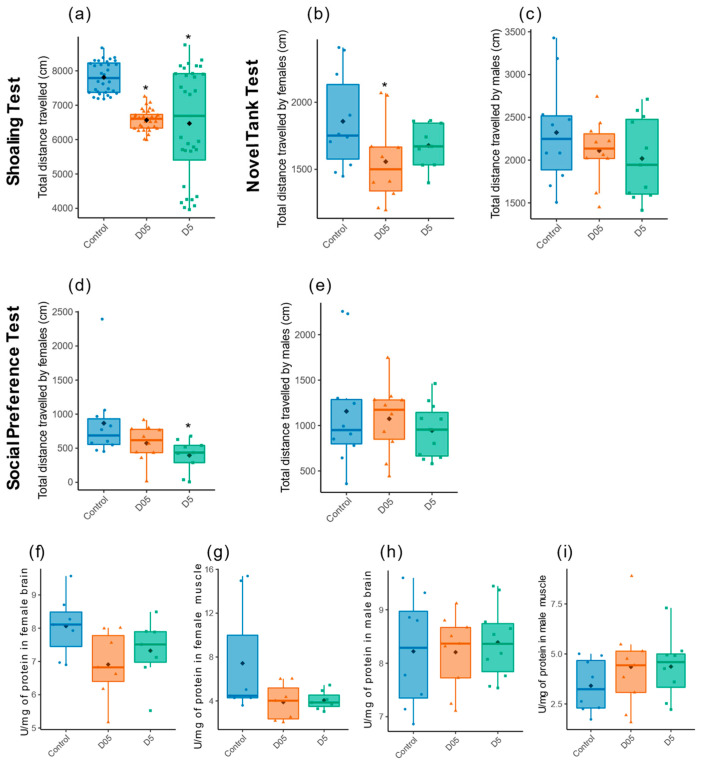
Effect of radiation exposure on locomotor activity and dosage of Acetylcholinesterase activity. Total distance travelled in centimeter (cm) was evaluated by (**a**) Shoaling test (*n* = 32), (**b**,**c**) Novel tank test and (**d**,**e**) Social preference test in females zebrafish (*n* = 9–10) and males (*n* = 10–11). Data are represented by mean (black point), median (horizontal line) and ±75th and 25th percentiles, and significance assessed by Gamma generalized linear mixed models (a) or Gaussian linear effects models (**b**–**e**). Measure of AChE activity in (**f**) brain and (**g**) muscle of females, and in males (**h**,**i**). Data are represented by mean (black point), median (horizontal line) and ±75th and 25th percentiles of 7 individuals per condition. AChE activity is shown as enzymatic activity Unit per mg of protein (U/mg protein). Statistical analysis was performed by using one-way ANOVA followed by Dunnet post-hoc test. * *p* < 0.05.

**Figure 4 cancers-14-03793-f004:**
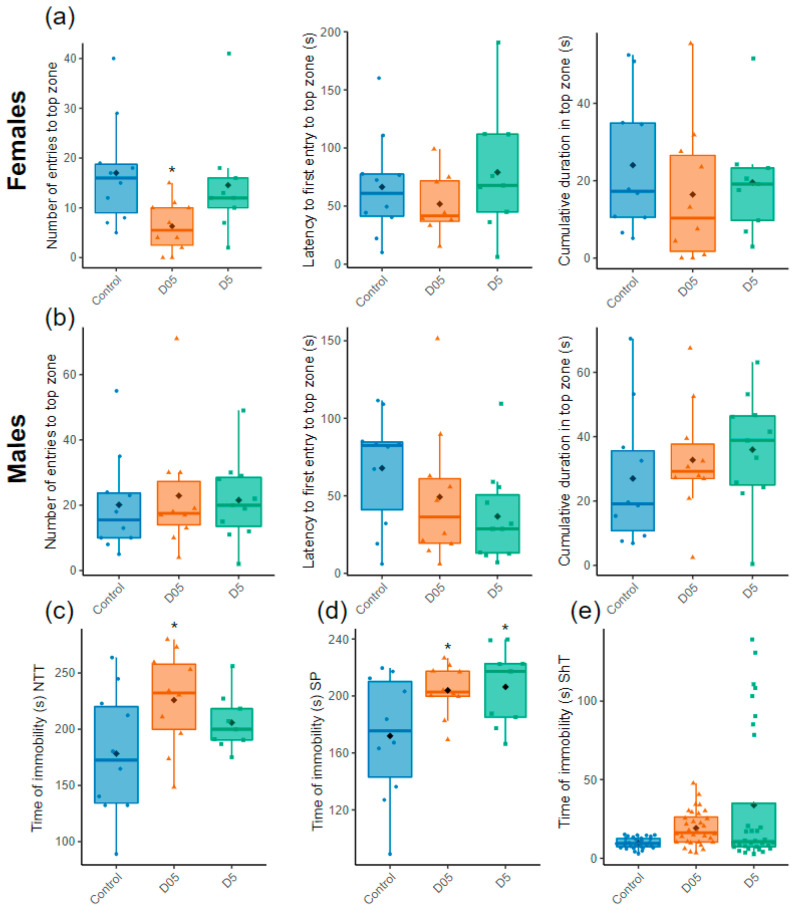
Assessment of stress behavior. (**a**) Number of entries, latency to first entry and time spent in the top zone in the novel tank test (NTT) for females and (**b**) males (*n* = 9–11). (**c**) Time of immobility of females in the NTT (*n* = 9–10), (**d**) the Social Preference test (SP) (*n* = 9–10 females or males) and (**e**) the Shoaling test (ShT) (*n* = 32 both females and males). Data are represented by mean (black point), median (horizontal line), ±75th and 25th percentiles and analyzed with negative binomial mixed effect models (**a**,**b**, left panels) or Gaussian linear effects models (**a**,**b**, middle and right panel, **c**,**d**) or Gamma generalized linear mixed models (**e**). * *p* < 0.05.

**Figure 5 cancers-14-03793-f005:**
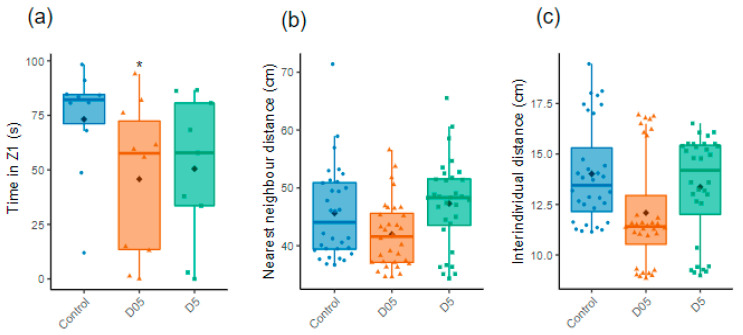
Assessment of individual and group social behavior. (**a**) Time in Z1 evaluated for females during the Social Preference test (*n* = 9–10). (**b**) Nearest neighbor distances and (**c**) Inter-individual distances analyzed during the Shoaling test (ShT) (*n* = 32). Data are represented by mean (black point), median (horizontal line), ±75th and 25th percentiles and analyzed with Gaussian linear effect model (**a**), Gamma generalized linear mixed models (**b**,**c**). * *p* < 0.05.

**Figure 6 cancers-14-03793-f006:**
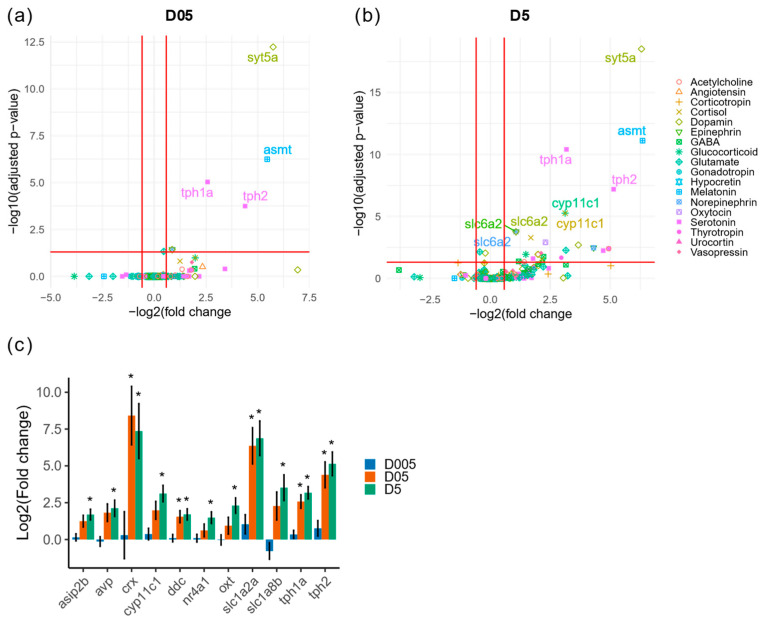
Transcriptomic analysis of genes with function in neurotransmission or neurohormones. (**a**) Volcano plot of differentially expressed genes with function in neurotransmission or brain neurohormones at D05 (0.5 mGy/h) and (**b**) at D5 (5 mGy/h). The biological process associated to each gene is indicated by the color and shape code. (**c**) Change in expression of dysregulated genes. * Indicate adjusted *p*-value (<0.05).

**Figure 7 cancers-14-03793-f007:**
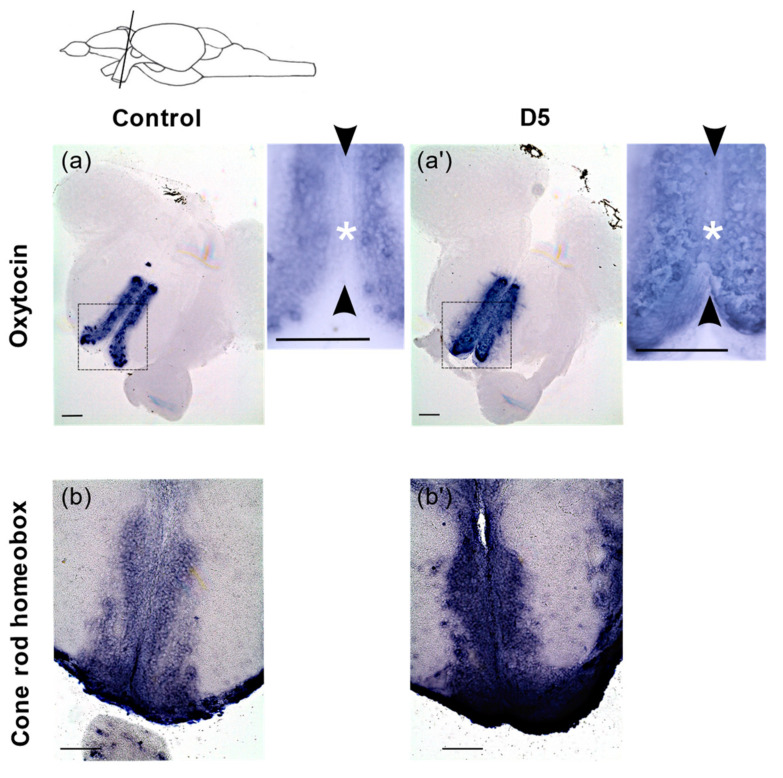
RNA in situ hybridization on adult brain sections showing expression of *oxt* and *crx* expressing cells. (**a**,**a**′) Coronal sections of brain showing expression of *oxt* in the anterior parvocellular preoptic nucleus (PPa) in control and 5mGy/(D5) irradiated fish. The black rectangle represents the area for the higher magnification images. (**b**,**b**′) Expression of *crx* in the PPa in the control and after irradiation at 5 mGy/h(D5). The black arrow heads delineate the ventricular zone (VZ) in the medial ventricular zone of the PPa and the white asterisk represents the location of VZ progenitors where ectopic expression is detected. Scale bar = 100 µm. The scheme at the top represents the location of the coronal sections in the brain.

## Data Availability

Transcriptomics data are available under the Gene Expression Omnibus GEO repository: GSE206573.

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
