# Peer review of "Revealing the Increased Stress Response Behavior through Transcriptomic Analysis of Adult Zebrafish Brain after Chronic Low to Moderate Dose Rates of Ionizing Radiation"

_cancers, 2022, doi:10.3390/cancers14153793_

Round 1
Reviewer 1 Report
This is a serious work pointing on possible connection between molecular mechanisms and behavioral effects of ionizing radiation.
However, I have one major concern regarding the statistical analysis. The authors choose (reasonably) the p-value threshold as 0.05 but do not mention the Bonferroni correction. Since multiple hypotheses (genes) were tested simultaneously, the above correction is crucial. If the Bonferroni correction has been performed, it should be clearly described. If not - unfortunately, the data analysis should be re-done.
Minor comments follow.
Abstract
line 31: Three behavioral tests evaluated locomotion, stress behavior and sociability response. - I suggest moving this piece downwards, before "Together,..."; Also, I'd expect short presentation of the behavioral results here (like in Conclusions).
Introduction
1) What is LD50 for zebrafish? Should be given for reference
2) The authors often use the term "dose" when it is actually "dose rate": e.g., 6mGy/h is dose rate
67: Refs. 18-20: The results of these studies are rather speculative. Very low exposures were involved. Confounding factors - e.g., mothers' stress - could not be taken into account quantitatively. Formulate more cautiously - e.g. "there is some evidence", "some researchers suggest", ...
69: Refs. 21-22: the same as above
73: risk of Parkinson disease was observed in a cohort of nuclear Russian workers - the same as above; Anyhow, citation needed
___________________________
Sect. 2.2 - add reference to Figure 2. It may be a good idea to split Fig. 2 into two - (a) experimental set-up and (b) results.
136: MCNP - software package? Please, clarify
198: dry ice - solid CO2? Then why 4 degrees? Or solid H2O? Then why "dry"?
line 233 and further: for clarity, I suggest using italics for software packages' names like RNA-STAR
244-246: present as a table
Figure 1
345: number of genes => percentage of genes?
Needs comment. 0.05 mGy/h: probably noise only - otherwise why no common genes with higher radiation doses? May be well related to non-application of Bonferroni correction. And probably also for most of 43 non-intersecting genes for 0.5 mGy/h.
Figure 2: insufficient quality!
____________________
720: chronic and low to moderate dose rates => chronic low to moderate dose rates ('and' seems unnecessary)
742: This research received no external funding - should be clarified. Cite the IRSN project number(s) or at least state that it was performed in the framework of the IRSN working program.
Author Response
- I have one major concern regarding the statistical analysis. The authors choose (reasonably) the p-value threshold as 0.05 but do not mention the Bonferroni correction. Since multiple hypotheses (genes) were tested simultaneously, the above correction is crucial. If the Bonferroni correction has been performed, it should be clearly described. If not - unfortunately, the data analysis should be re-done.
We want to thank the reviewer for this question. We confirm that we corrected the p value for multiple testing. The criterion of selection is not based on p-value but on adjusted p-value corrected by the Benjamini and Hochberg method, also called false discovery rate (fdr). Several methods are valid for correcting p-value for multiple testing, including the fdr and the method of Bonferroni. Other exist including the method of Benjamini-Yekutieli. The false discovery rate (fdr) is thus a valid method to adjust p-value for transcriptomic data and is commonly used in the field with over 87000 citations (including recent publications) in Pubmed. We added the appropriate reference in the material and Method line 224.
Abstract
- line 31: Three behavioral tests evaluated locomotion, stress behavior and sociability response. - I suggest moving this piece downwards, before "Together,..."; Also, I'd expect short presentation of the behavioral results here (like in Conclusions).
We made these changes as requested line 36
Introduction
- 1) What is LD50 for zebrafish? Should be given for reference
Thank you for this question. The LD50 for ionizing radiation for zebrafish is derived from acute exposure to high dose of X-rays. For early embryos it is between 1 and 5 Gy (Cell Biol. Int. 2019, 43, 516–527). For adults, mortality starts to be observed at 20Gy due to intestinal and hematopoietic damages (https://journals.sagepub.com/doi/pdf/10.1016/j.icrp.2009.04.011). There is now documented LD50 for chronic radiation. However, we don’t think this value is valuable in the context of our study on chronic low dose radiation.
- 2) The authors often use the term "dose" when it is actually "dose rate": e.g., 6mGy/h is dose rate
We completely agree with this comment, we made the changes accordingly through the manuscript
- 67: Refs. 18-20: The results of these studies are rather speculative. Very low exposures were involved. Confounding factors - e.g., mothers' stress - could not be taken into account quantitatively. Formulate more cautiously - e.g. "there is some evidence", "some researchers suggest",
We tempered this sentence accordingly line 66
- 69: Refs. 21-22: the same as above
We tempered this sentence accordingly line 68
- 73: risk of Parkinson disease was observed in a cohort of nuclear Russian workers - the same as above; Anyhow, citation needed
Thank you for this comment, the reference was added
Results
- 2.2 - add reference to Figure 2. It may be a good idea to split Fig. 2 into two - (a) experimental set-up and (b) results.
The figure was modified accordingly. For a better visibility we split the Figure 2 into two different figures and made the appropriate changes. We also added references.
- 136: MCNP - software package? Please, clarify
The version of the of the software was added in the material and Methods
- 198: dry ice - solid CO2? Then why 4 degrees? Or solid H2O? Then why "dry"?
The dry ice is installed at the top of the Precellys apparatus to cool the temperature at 4°C and avoid increased temperature due to bead beating. To clarify this point we modified the sentence line 188
- line 233 and further: for clarity, I suggest using italics for software packages' names like RNA-STAR
Changes were made
- 244-246: present as a table
Done as a supplementary table S2
- Figure 1: 345: number of genes => percentage of genes?
This information is conveyed in the core of the Figure 1 by the dot plot size which corresponds to the gene ratio. As this information is already indicated, we didn’t duplicate it in the figure to avoid redundancy, but modified the legend to explain this point.
- Figure 2: insufficient quality!
This figure is now split into two separated figures to improve readability
- 720: chronic and low to moderate dose rates => chronic low to moderate dose rates ('and' seems unnecessary)
Modification was done
- 742: This research received no external funding- should be clarified. Cite the IRSN project number(s) or at least state that it was performed in the framework of the IRSN working program.
Information was added line 765
Reviewer 2 Report
The manuscript of Cantabella et al “Revealing the increased stress response behavior through transcriptomic analysis of adult zebrafish brain after chronic and low to moderate dose rates of ionizing radiation” report the identification of dose-dependent increase of differentially expressed genes involved in neurotransmission, neuro-hormones and hypothalamic-pituitary-interrenal axis functions in irradiated zebrafish, suggesting their involvement in the anxiety-like behavior observed. Several points need to be clarified:
1-the signal of the in situ hybridization presented in fig 6, panels a and a’ need to be quantified. It is particularly important to show that with a similar signal between control and irradiated animal, the size of the signal is extended in this later; this is not clear in the figure since the sections shown are not identical between the two fishes. Also, to demonstrate an increase of the signal in the irradiated animals, a control in situ should be performed with an invariant gene on the same section to compare with oxt signal.
2-since the increase in gene expression is the presumed basis for the differences in stress-behavior observed, such increase at the protein level should be shown (for crx for exemple).
3-since only females showed a significant stress behavior, the authors should show that the weight between males and females are identical, otherwise an increased irradiation in the female (if their weight is reduced) could provide an explanation for the difference in stress response.
4-it would be very interesting to know if some senescence event at the cellular level are induced in irradiated animals.
CRF in lane 86 should be explained
Author Response
- 1-the signal of the in situ hybridization presented in fig 6, panels a and a’ need to be quantified. It is particularly important to show that with a similar signal between control and irradiated animal, the size of the signal is extended in this later; this is not clear in the figure since the sections shown are not identical between the two fishes. Also, to demonstrate an increase of the signal in the irradiated animals, a control in situ should be performed with an invariant gene on the same section to compare with oxt signal.
Thank you very much for this question. We agree with this reviewer that RNA in situ hybridization is a semi quantitative method. The point we want to raise is that cells lining the ventricular zone do not express oxt in the control situation but rather express oxt in cells located more laterally. While after irradiation at 5 mGy/h we do detect oxt positive cells in the ventricular zone and more laterally. This is thus a ‘all or none response’ (in the ventricular zone), which is well conveyed by the usage of the term ‘ectopic’ in our study. To improve this point, we added this information line 556 and modified the Figure 7 (former Figure 6). In addition, to please this reviewer, we now show the quantitative expression of oxt by fluorescent in situ and confirm the statistical upregulation of oxt. This novel data is presented in Supplementary figure S1. For the last part of the reviewer comments, we do not believe that showing in situ for non-differentially expressed gene is appropriate. The concept of changes compared to constant expression (housekeeping genes) applies to Western Blot and Q RT PCR, but not for RNA in situ hybridization. In the case of oxt expression, we compare a zone were cells express oxt ectopically. This result was repeated 3 times on independent brains as described. We thus do not need to show expression of a house keeping gene, as we have in this case a ‘yes or no’ result. In addition, showing the expression of another gene with the same domain of expression as oxt in the brain is not possible. In this context it is widely acknowledge that it is appropriate to show in situ pattern as we present it in the present study: same area of the brain from individual exposed to different conditions (here radiation), and in replicate. We show transcriptional upregulation by RNAseq, classical in situ and fluorescent in situ. We believe that using 3 independent techniques for demonstrating the validity of our results (at the transcriptional level) is enough.
- 2-since the increase in gene expression is the presumed basis for the differences in stress-behavior observed, such increase at the protein level should be shown (for crx for exemple).
Thank you for this comment. The aim of this study is to use gene expression as a way to identify possible mechanisms linked to stress behaviour. We thus use a global approach with transcriptomics data to dissect the possible molecular mechanisms, as announced in the title of our manuscript. Confirming the differences at the protein levels is thus interesting but out of the scope of the present study which focuses on the global transcriptomic responses. To clearly state this point we added a sentence lines 635-636 in the paper.
- 3-since only females showed a significant stress behavior, the authors should show that the weight between males and females are identical, otherwise an increased irradiation in the female (if their weight is reduced) could provide an explanation for the difference in stress response.
We tested if males and female had different weight and we didn’t detect significant difference. We added a sentence lines 133-134
- 4-it would be very interesting to know if some senescence event at the cellular level are induced in irradiated animals.
This is a very interesting point, thank you for asking about senescence and indeed cellular senescence is induced by high dose of radiation. We checked the differential expression of several marker of cellular senescence including P53, CDKN2A and CDKN2B, but these genes were not deregulated. We also didn’t find enrichment of GO term related to senescence. Increased apoptosis, DNA damages and oxidative stress are processes implicated in cellular senescence (Trends in Cell Biology, June 2018, Vol. 28, No. 6), and we didn’t observe any significant changes (line 616 onward). This suggests that chronic low dose radiation do not induce the same response compared to high dose. We added our data on cellular senescence line 616 in order to be comprehensive.
- CRF in lane 86 should be explained
Modification was made
Reviewer 3 Report
Dear Authors,
my very best compliments on your very interesting manuscript.
I do not have comments/questions to ask you except a short one like the following:
at lines 73-74.
do you have a literature source supporting the following statement of yours: "For instance, an elevated risk of Parkinson disease was observed in a cohort of nuclear Russian workers after chronic exposure to external .-ray and neutrons"?
good luck for publication
Author Response
Thank you for your comments and your interest in this work. We added the lacking reference.
Round 2
Reviewer 1 Report
Good luck!
Reviewer 2 Report
the authors have modified the manuscrpt and it could now be accepted for publication.